# Genome-Wide Investigation and Functional Verification of the ZIP Family Transporters in Wild Emmer Wheat

**DOI:** 10.3390/ijms23052866

**Published:** 2022-03-05

**Authors:** Fangyi Gong, Tiangang Qi, Yanling Hu, Yarong Jin, Jia Liu, Wenyang Wang, Jingshu He, Bin Tu, Tao Zhang, Bo Jiang, Yi Wang, Lianquan Zhang, Youliang Zheng, Dengcai Liu, Lin Huang, Bihua Wu

**Affiliations:** 1State Key Laboratory of Crop Gene Exploration and Utilization in Southwest China, Chengdu 611130, China; fangyigong2021@126.com (F.G.); 17863660603@163.com (T.Q.); hylsichuan@163.com (Y.H.); 13438830038@163.com (Y.J.); jialiu251776931@163.com (J.L.); cd_wangwy@163.com (W.W.); hejingshu@sicau.edu.cn (J.H.); jiangbo19880229@163.com (B.J.); wangyi@sicau.edu.cn (Y.W.); zhanglianquan1977@126.com (L.Z.); ylzheng@sicau.edu.cn (Y.Z.); dcliu7@sicau.edu.cn (D.L.); 2Triticeae Research Institute, Sichuan Agricultural University, Chengdu 611130, China; 3Rice Research Institute, Sichuan Agricultural University, Chengdu 611130, China; tubin14216@sicau.edu.cn (B.T.); zhangtaomx@163.com (T.Z.)

**Keywords:** ZIP gene family, wild emmer wheat, yeast complementation, transgenic verification

## Abstract

The zinc/iron-regulated transporter-like protein (ZIP) family has a crucial role in Zn homeostasis of plants. Although the ZIP genes have been systematically studied in many plant species, the significance of this family in wild emmer wheat (*Triticum turgidum* ssp. *dicoccoides*) is not yet well understood. In this study, a genome-wide investigation of ZIPs genes based on the wild emmer reference genome was conducted, and 33 *TdZIP* genes were identified. Protein structure analysis revealed that TdZIP proteins had 1 to 13 transmembrane (TM) domains and most of them were predicted to be located on the plasma membrane. These *TdZIPs* can be classified into three clades in a phylogenetic tree. They were annotated as being involved in inorganic ion transport and metabolism. Cis-acting analysis showed that several elements were involved in hormone, stresses, grain-filling, and plant development. Expression pattern analysis indicated that *TdZIP* genes were highly expressed in different tissues. *TdZIP* genes showed different expression patterns in response to Zn deficiency and that 11 genes were significantly induced in either roots or both roots and shoots of Zn-deficient plants. Yeast complementation analysis showed that *TdZIP1A-3*, *TdZIP6B-1*, *TdZIP6B-2*, *TdZIP7A-3*, and *TdZIP7B-2* have the capacity to transport Zn. Overexpression of *TdZIP6B-1* in rice showed increased Zn concentration in roots compared with wild-type plants. The expression levels of *TdZIP6B-1* in transgenic rice were upregulated in normal Zn concentration compared to that of no Zn. This work provides a comprehensive understanding of the ZIP gene family in wild emmer wheat and paves the way for future functional analysis and genetic improvement of Zn deficiency tolerance in wheat.

## 1. Introduction

Zinc (Zn) is one of the most essential micronutrients for plants and humans and plays a critical role in diverse biochemical processes [1,2]. Zn is irreplaceable for plant normal growth and development [3]. Both deficient and excess Zn has negative effects on the physiological and biochemical processes of the plant [4]. The Zn availability in agriculture soils of many parts of the world is low, which leads to yield reduction and poor nutritional quality in harvested grains [5]. It was estimated that one-third of the world’s population suffers from inadequate intake of Zn, resulting in various health problems [5,6,7]. A food-based strategy (biofortification) is considered the most cost effective and sustainable option for improvement of human health [8]. Thus, improving the Zn nutrition in crop varieties is an important goal of both public and private breeding programs to overcome the Zn deficiency problem.

To adapt low and fluctuating Zn availability in soil, plants have developed various mechanisms to equipoise the uptake, utilization, and storage of Zn ion. Three metal transporter families, namely, the Zn/iron-regulated transporter-like protein (ZIP), the cation diffusion facilitator (CDF), and the heavy metal ATPase (HMA) have been shown to be involved in Zn homeostasis of plants [9]. Among them, the ZIP family has been validated to be involved in the Zn^2+^ uptake from soil and the transport of this metal from extracellular or intracellular compartment into the cytoplasm [10].

Several ZIP family transporters have been identified from various plant species [2,11,12]. Most of the ZIP proteins have eight transmembrane (TM) domains and similar membrane topologies. Many members have relatively long variable regions between TM regions III and IV, which underlies the length variations (300 to 480 amino acid residues) of different ZIP proteins. This variable region is histidine-rich and resided in the cytoplasm, which is predicted to be involved in the binding and the transport of metal ions [13,14].

In plants, multiple members of the ZIP family have been shown to respond to Zn deficiency, and some of the Zn deficiency-inducible ZIP genes reported thus far have the ability to transport Zn [12,15,16,17]. For example, the *AtZIP1* and *AtZIP2* genes played an important role in Mn and Zn translocation from the roots to the shoots in *Arabidopsis thaliana* [18]. In barley, six *HvZIP* genes were highly induced by Zn deficiency, which is associated with enhanced uptake and root-to-shoot translocation of Zn under low Zn conditions [8]. The *OsZIP1*, *OsZIP3*, *OsZIP4*, *OsZIP5*, and *OsZIP8* were shown to transport of Zn and three of them (*OsZIP4*, *OsZIP5*, and *OsZIP8*) were involved in response to Zn-deficiency in rice [12,16,19,20]. Evens et al. (2017) identified 14 ZIPs in hexaploid wheat and verified five members (*TaZIP3*, *TaZIP6*, *TaZIP7*, *TaZIP9*, and *TaZIP13*) as functional Zn transporters [21].

Wild emmer wheat (*Triticum turgidum* ssp. *dicoccoides*, 2n = 4× = 28, AABB), the tetraploid progenitor of domesticated wheat, represents an important genetic source for high-grain Zn, Fe, and protein concentrations [5,22,23,24]. Although wide variation in response to Zn deficiency has been reported in wild emmer, little is known about the ZIP family genes except for *TdZIP1*, a Zn transporter with higher expression under Zn deficiency, which can complement the function of the *zrt1*/*zrt2* yeast mutant [25]. In the present study, 33 *TdZIP* genes were identified from the wild emmer wheat genome. The gene structure, expression profiles induced by Zn deficiency, and Zn transport capability of these *TdZIP* genes were systematically analyzed. The function of *TdZP6B-1* was further verified by transgenic rice. The results will improve our understanding of the regulatory roles of *TdZIPs* in Zn accumulation and will lay a foundation for further functional study and genetic improvement of Zn deficiency tolerance in wheat.

## 2. Results

### 2.1. Phylogenetic Relationships and Comparative Analysis of the ZIP Gene Family in Wild Emmer Wheat

A total of 33 putative ZIP genes were identified and confirmed from the wild emmer reference genome. These ZIP genes were tentatively designated as *TdZIP1A-1* to *TdZIP7B-3* (Table 1) according to their locations on chromosomes. We found that these ZIPs were unevenly distributed on the chromosomes, with 16 and 17 genes positioned on the A and B chromosomes, respectively (Table 1, Appendix A). The first chromosome group had the largest number of *TdZIPs* (10 genes), followed by the second (7 genes), the sixth (5 genes), and the seventh chromosome groups (5 genes). The remaining chromosomes had one *TdZIP* gene each (Appendix A).

Gene structure analysis revealed that the number of introns varied from 0 to 11 (Appendix A), and exons ranged from 1 to 12 among the 33 *TdZIP* genes. The amino acid lengths of the *TdZIPs* varied from 68 (TdZIP2A-2) to 577 (TdZIP1A-3 and TdZIP1B-3), the PIs ranged from 5.05 *(TdZIP2A-2*) to 9.8 (TdZIP2A-3), and these proteins had 1 to 13 TM domains. Using the MEME tool, we identified 15 conserved motifs with length varied from 6 to 50 amino acids (Appendix A). Most of the TdZIP proteins were predicted to be localized on the plasma membrane, except three TdZIPs (TdZIP6A-2, TdZIP6B-2, and TdZIP7B-1) were predicted to be localized on the endoplasmic reticulum (ER) (Table 1), suggesting that these three genes may be responsible for transporting Zn from the ER to the cytoplasm.

We constructed a phylogenetic tree using 33 *TdZIPs* and 29 ZIPs from rice, maize, and Arabidopsis to further study the phylogenetic relationships between *TdZIPs* and other ZIPs in plants. The result is presented in Figure 1. In the phylogenetic tree, these ZIPs were divided into three clades: clade I (1 *OsZIP*, 2 *ZmZIPs*, 2 *AtZIPs*, and 6 *TdZIPs*), clade II (5 OsZIPs, 2 ZmZIPs, 4 AtZIPs, and 13 TdZIPs), and clade III (5 OsZIPs, 3 ZmZIPs, 5 *AtZIPs*, and 14 *TdZIPs*). We further investigated the collinearity relationship of ZIP genes between wild emmer wheat and common wheat. The 33 *TdZIPs* genes were blasted against the genome of Chinese Spring (CS). Among them, 32 ZIPs had homologous genes/sequences in the CS genome, and 15 genes were able to be detected with 82–100% identities. One gene *TdZIP5B-1* was not matched with CS genes (Appendix A, Appendix A).

### 2.2. Functional Annotation and Promoter Analysis of TdZIP Genes

To investigate the potential functions of ZIPs, we performed GO, KOG, and Swiss_Prot_annotation analyses for the 33 *TdZIPs*. GO terms for those ZIP genes can be classified into three categories: biological process (BP), cellular component (CC), and molecular function (MF). In the BP annotation, TdZIP proteins predicted their functions in cellular process (GO:0009987), single-organism process (GO:44699), location (GO:0051179), and response to stimulus (GO:0050896). In the CC annotation, TdZIP proteins percentage annotated with the membrane (GO:0016020), membrane part (GO:0044425) and cell (GO:0005623), cell part (GO:0044464), organelle (GO:0043226), and organelle part (GO:0044422). Transporter activity (GO:0005215), catalytic activity (GO:0003824), and binding (GO:0005488) were annotated in the MF annotation (Appendix A). KOG annotation revealed that all *TdZIPs* were involved in inorganic ion transport and metabolism (Appendix A). In addition, Swiss_Prot_annotation showed that most of *TdZIPs* (27 genes) were orthologous to *OsZIP* genes in rice (Appendix A). The remaining *TdZIPs* were orthologous to *AtZIP* genes in *Arabidopsis thaliana*. These results were consistent with the phylogenetic analysis data.

To further understand the role of *TdZIPs*, we used the 2000 bp upstream sequence from the translation initiation site (ATG) of *TdZIP* genes and analyzed in the PlanCARE for cis-element prediction. The results showed that hormone, stresses, and grain-filling, and developmental responsive cis-elements were widely predicted in *TdZIPs* promoters. In hormone category, abscisic acid (ABRE), MeJA (CGTCA_motif), auxin (TGA_element), salicylic acid (TCA), and gibberellin (GARE_motif) responsive cis-elements were dominant (Appendix A). Abscisic acid-responsive element was the most frequent in *TdZIPs* promoters. Category for stresses had a different type of cis-regulatory elements. For instance, LTR involved in low-temperature responses, MBS in drought-inducibility, ARE in anaerobic induction, and TC_rich_repeats in defense and stress responses (Appendix A). In grain-filling and developmental category, zein metabolism regulation (O2_site) was the most frequent elements in *TdZIP* promoters, followed by meristem expression (CAT_box), light responsive element (TCCC_motif), seed-specific regulation (RY_element), and endosperm expression (GCN4_motif). These results showed that the *TdZIPs* may undergo different types of transcriptional regulation and the potential diversity functions of *TdZIPs* in wild emmer.

### 2.3. Expression Pattern of TdZIP Genes under Zn-Deficient Stress

To understand the potential expression pattern of *TdZIP* genes in different tissues, we retrieved the expressions of 33 *TaZIPs* in different tissues sampled at different time points from public available RNA-seq data of wild emmer [26]. The log_2_ (TPM + 1) value was used for the heat map display (Appendix A). We found *TdZIP* genes showed different expression patterns. According to the expression data, 14 *TdZIP* genes, namely, *TdZIP1A-1*, *TdZIP1A-2*, *TdZIP1A-3*, *TdZIP1B-3*, *TdZIP1B-5*, *TdZIP2A-2*, *TdZIP2A-3*, *TdZIP2B-3*, *TdZIP6A-2*, *TdZIP6B-1*, *TdZIP6B-2*, *TdZIP6B-3*, *TdZIP7A-2*, and *TdZIP7B-2*, were highly expressed in different tissues sampled at different time points. Some genes were expressed in all sampled tissues. For example, *TdZIP6B-2* had high expression in root, leaf, developing spike, lemma, glume, flower, and grain at most time points. *TdZIP1A-1* was highly expressed in leaf, lemma, glume, and grain. In addition, some genes only expressed at specific developmental stages of specific tissues. For example, four genes—*TdZIP1A-2*, *TdZIP1A-3*, *TdZIP1B-3*, and *TdZIP6B-2*—were highly expressed in roots at 20 days from sowing (DFS). *TdZIP2A-3* showed high expression levels at developing stages of spike length 2.5, 4.5, and 5.5 cm as well as in lemma and glume at 112 DFS. *TdZIP6B-2* was extremely highly expressed in flower at 105 and 110 DFS. Seven genes (*TdZIP1A-1*, *TdZIP1A-3*, *TdZIP1B-3*, *TdZIP2A-4*, *TdZIP6A-2*, *TdZIP6B-2*, and *TdZIP6B-3*) showed high expression levels in grain at 123 DFS (Appendix A).

To further investigate the transcript changes of *TdZIPs* in response to different concentrations of ZnSO_4_ treatments, we selected 14 *TdZIPs* represent the three clades of *TdZIPs* in a phylogenetic tree (Figure 1), and their expression patterns in roots and leaves were analyzed by qRT-PCR under Zn-deficient and normal Zn concentrations (Appendix A). Three expression patterns were observed: (i) transcript levels of nine genes (*TdZIP1A-2*, *TdZIP1A-3*, *TdZIP1B-5*, *TdZIP2B-3*, *TdZIP4A-1*, *TdZIP7A-2*, *TdZIP7A-3*, *TdZIP7B-2*, and *TdZIP7B-3*) were significantly upregulated in roots of Zn-deficient plants, while not in leaves; (ii) the expression levels of three genes (*TdZIP1A-4*, *TdZIP6B-1*, and *TdZIP7B-1*) were significantly upregulated in both roots and shoots of Zn-deficient plants; and (iii) three genes (*TdZIP1B-3*, *TdZIP6A-2*, and *TdZIP6B-2*) in both leaves and roots showed little or no response to Zn deficiency or normal Zn concentration (Figure 2).

### 2.4. Yeast Complementation Analysis

To determine the biological function of *TdZIPs*, we chose six *TdZIP* genes (*TdZIP1A-2*, *TdZIP1A-3*, *TdZIP6B-1*, *TdZIP6B-2*, *TdZIP7A-3*, and *TdZIP7B-2*) for yeast complementation analysis. The coding sequences of these genes were cloned and expressed in the wild-type yeast strain DY1457 and extracellular Zn transporter (*zrt1*/*zrt2*) double mutant. The *zrt1*/*zrt2* mutant cells transformed with *TdZIP* genes were grown on SD media containing different Zn concentrations. Among them, cells transformed with *TdZIP6B-1* had the strongest growth on SD media with ZnSO_4_. *TdZIP* genes *TdZIP1A-2*, *TdZIP1A-3*, *TdZIP6B-1*, and *TdZIP7B-2* were able to fully complement and *TdZIP6B-2* and *TdZIP7A-3* were able to partially complement the growth phenotypes of the double *zrt1*/*zrt2*Δ yeast mutant (Figure 3). These results suggest that the six *TdZIPs* genes have the ability to transport Zn.

### 2.5. Analysis of Phenotypes and Metal Concentrations in TdZIP6B-1 Overexpression Rice Plants

The yeast complementation experiment showed that *TdZIP6B-1* has the highest ability to complement the growth phenotypes of the double *zrt1*/*zrt2*Δ yeast mutant. The cDNA sequence of *TdZIP6B-1* shares 95.77% identity with that of *TaZIP6B-1* in CS. *TdZIP6B-1* was phylogenetically close to Arabidopsis gene *AtZIP11*, maize gene *ZmZIP2*, and rice gene *OsZIP2* (Figure 1). We constructed transgenic plants overexpression *TdZIP6B-1* to further verify the function of *TdZIP6B-1*. The *TdZIP6B**-1* overexpression line (TdZIP6B) together with wild-type (WT) cultivar (*Oryza. Sativa* L. spp. *Japonica*) were exposed to no Zn (0 mg/L ZnSO_4_) and normal Zn conditions (8.6 mg/L ZnSO_4_) (Figure 4A). The *TdZIP6B**-1* overexpression lines had higher fresh weight of roots and shoots than those of WT plants after 14 days growth under normal Zn conditions (Figure 5B, C). The total roots length of overexpression lines was significantly longer than that of WT plants (Figure 5A,D). The total roots area and volume were higher than those of WT plants, but the differences were not significant (Figure 5E,F). In roots, the expression levels of *TdZIP6B-1* were significantly higher in normal Zn condition than those of in no Zn condition at 5 days post-treatment (Figure 4B). No significant differences were observed for the expression levels of *TdZIP6B-1* at 7 and 9 days. In shoots, the expression levels of *TdZIP6B-1* were significantly higher in normal Zn condition than those of in no Zn condition at 7 days post-treatment (Figure 4C). We further examined the Zn, Mn, and Fe concentrations in roots and shoots after 14 days of growth under normal Zn condition. The results showed that the transgenic plants had higher Zn, Fe, and Mn concentrations compared with that of WT plants in roots (Figure 4D), while those of in shoots between overexpression lines and WT plants were not significantly different (Figure 4E,F).

## 3. Discussion

Zn is a micronutrient element that is necessary for all living organisms [8,27]. Zn deficiency in major food crops could lead to yield reduction and poor nutritional quality [2]. Zn homeostasis is tightly regulated by kinds of proteins, with Zn transporter proteins being particularly important [27,28]. The ZIP transporters contribution to Zn homeostasis has been widely reported in many plant species, including common wheat [21]. To our knowledge, however, the ZIP family has not been well studied in wild emmer wheat. In the present study, we performed a genome-wide investigation of ZIP family in wild emmer and identified 33 *TdZIP* genes. Although most (97%) *TdZIPs* had homologous genes/sequences in common wheat cv. CS genome, large sequences differences in ZIP genes were detected between wild emmer and CS. These *TdZIP* genes in wild emmer can provide new candidate genes for improving the nutritional quality of wheat.

The phylogenetic tree of the *TdZIPs* genes was clustered into three clades. In clade I, six *TdZIPs* were closely related to *OsZIP10*, *AtZIP4*, *AtZIP9*, *ZmZIP5*, and *ZmZIP7*. The *OsZIP10* was found to be associated with grain Zn content in rice [20]. A previous study had shown that *AtZIP9* complemented a yeast Mn uptake-deficient mutant, while it did not complement the Zn uptake-deficient mutant [29]. *ZmZIP5* played an important role in Zn and Fe uptake and root-to-shoot translocation in maize [30]. In clade II, 13 *TdZIPs* were clustered with *OsZIP1*, *AtZIP2*, and *AtZIP11*, which were previously shown to have ability to transport Zn and response to Zn deficiency [21,29,31]. A total of 14 *TdZIPs* in clade III were clustered together with *OsZIP3*, *OsZIP4*, *OsZIP5*, *OsZIP8*, *OsZIP9*, *ZmZIP3*, *ZmZIP4*, *ZmZIP8*, *AtZIP1*, *AtZIP3*, *AtZIP5*, *AtZIP7*, and *AtZIP12*. The *OsZIP3*, *OsZIP4*, *OsZIP5*, *OsZIP8*, *AtZIP1*, *AtZIP3*, *AtZIP7*, and *AtZIP12* had been verified to have ability to transport Zn in plants [11,12,20,29,32,33], while the *ZmZIPs* could complement the transport of Zn and Fe in yeast mutants [34]. Taken together, these data indicate that the *TdZIPs* closely related to *OsZIP*, *ZmZIP*, and *AtZIP* genes are likely to have the ability to absorb and transport Zn and Mn and respond to Zn deficiency.

The cis-regulatory elements present in the promoter region have an important role in plant regulation control and in different stimulus responsive genes [35]. On the basis of cis-element analysis, many *TdZIPs* were predicted to have hormone, stress, grain-filling, and developmentally responsive cis-elements. We have identified different types of cis-elements such as the hormones ABRE, which participates in ABA responses [36]; CGTCA_motif, which has a role in MeJA responses [37]; and TGA_element, which participates in auxin responses [38]. The cis-element for stress responses is LTR, which is involved in low-temperature responses [39]; meanwhile, MBS functions in drought-inducibility [40], ARE is involved in anaerobic induction, and TC_rich_repeats participate in defense and stress responses [41]. For grain-filling and developmental, O2_site has a role in zein metabolism regulation [42], CAT_box functions in meristem expression [38], TCCC_motif participates in light responsiveness [43], RY_element functions in seed-specific regulation [44], and GCN4_motif is involved in endosperm expression [45]. The presence of different types of cis-elements in *TdZIPs* implies the different transcriptional regulatory mechanisms in which the *TdZIPs* genes may be involved.

The expression pattern of genes is often tightly correlated with their functions [30]. The expression data of *TdZIPs* in different tissues at different time points are important for understanding genes’ functions in which they participate. Previous studies have reported that ZIP genes were mainly expressed in roots and shoots of rice, Arabidopsis, and common wheat [29,34,46,47]. In this study, six TdZIPs (*TdZIP1A-1*, *TdZIP1A-2*, *TdZIP1A-3*, *TdZIP1B-3*, *TdZIP6B-2*, and *TdZIP7B-2*) were mainly expressed in root and/or leaf tissues, suggesting their functions in these tissues, which is consistent with previous studies [8,46]. We found some *TdZIP* genes were highly expressed in developing spike and glume (e.g., *TdZIP1B-5*, *TdZIP2B-3*, and *TdZIP2A-3*), flower (*TdZIP6B-3*), and grain (*TdZIP1A-1* and *TdZIP6B-2*), suggesting their functions in spike developmental and grain filling processes. Interestingly, *TdZIP6B-2* had high expression in most tissues sampled at different time points, implying the potential diverse functions of this gene.

Several ZIP genes related to Zn transportation were upregulated in different tissues under Zn deficiency. For example, *OsZIP1* was upregulated in roots and the transcripts were not detected in shoots under Zn deficiency [11,32,46,48]. Zn deficiency induced the expression of *HvZIP6* in roots, while it did not in shoots [8]. *TaZIP3*, *TaZIP5*, *TaZIP6*, *TaZIP7*, and *TaZIP13* were upregulated in both roots and shoots under Zn deficiency [21]. In the present study, different expression patterns of *TdZIPs* in roots and shoots responding to Zn deficiency were determined. We observed that 11 *TdZIPs* were significantly induced in roots under Zn deficiency, of which three genes (*TdZIP1A-4*, *TdZIP6B-1*, and *TdZIP7B-1*) were highly induced in both roots and shoots of Zn-deficient plants, suggesting that these genes are involved in Zn deficiency responses and enhanced absorbance and root-to-shoot translocation of Zn. Three genes (*TdZIP1B-3*, *TdZIP6A-2*, and *TdZIP6B-2*) showed little or no sensitivity to Zn deficiency or the normal Zn concentration, suggesting that they contribute little to the uptake and root-to-shoot translocation of Zn.

High-affinity Zn transporter *Zrt1* is required for yeast growth in Zn-limiting media. Low-affinity transporter *Zrt2* can transport Zn under mild or abundant Zn concentrations, whereas it cannot transport Zn under more severe Zn-limiting media [49,50]. Mutations on each of them will lead to yeast growth abnormal under Zn deficiency. Therefore, the yeast *zrt1*/*zrt2*Δ mutant has been widely used to demonstrate the complement ability of Zn transporter genes in rice, maize, barely, Arabidopsis, and common wheat under Zn-deficient conditions [18,19,21,51]. In wheat, *TdZIP1* and *TaZIP3*, *-5*, *-6*, *-7*, and *-13* have been identified to have the ability to transport Zn by functional complementation study of the *zrt1*/*zrt2* yeast mutant [21,25]. In this study, six genes that belong to three clades of *TdZIPs* (Figure 3) were randomly selected to rescue the growth defects in the zrt1/zrt2Δ mutant under Zn-deficient conditions. The results revealed that four genes *TdZIP1A-2*, *TdZIP1A-3*, *TdZIP6B-1*, and *TdZIP7B-2* rescued the growth defects of the zrt1/zrt2Δ mutant, suggesting that these *TdZIPs* could transport Zn effectively.

In the present study, we further verified the Zn transport capability of *TdZIP6B-1* by rice transformation. We found that the expression levels of *TdZIP6B-1* in transgenic rice lines were higher under normal Zn condition than that of no Zn condition in both the roots and shoots. The Zn concentrations in roots of *TdZIP6B-1* overexpression lines were higher than the WT plants. Previous reports showed that the overexpression of *OsZIP4* and *OsZIP5* in rice [12,52] and *TaZIP13-B* [53] in *Arabidopsis* could enhance the concentration of Zn in roots and shoots. Our preliminary results verified that the *TdZIP6B-1* has the ability to uptake and probably induce root-to-shoot translocation of Zn. In addition, the accumulated Fe and Mn concentrations in roots and shoots of overexpression lines indicate that *TdZIP6B-1* may have pleiotropic effects on metal ion absorption in plants.

## 4. Materials and Methods

### 4.1. Identification and Bioinformatic Analysis of TdZIP Genes

The wild emmer wheat genome sequences (Triticum_dicoccoides.WEWSeq_v.1.0.dna.toplevel.fa.gz) and annotation profiles (Triticum_dicoccoides.WEWSeq_v.1.0.50.gff3.gz) [26] were downloaded from Ensembl Plant database (ftp://ftp.ensemblgenomes.org/pub/plants/release-50/fasta/triticum_dicoccoides/, accessed on 10 January 2022). The ZIP DNA-binding domain (PF02535.24) downloaded from Pfam protein families database (http://pfam.xfam.org/, accessed on 10 January 2022) was used to identify ZIP genes from the wild emmer genome using HMMER 3.0. The TdZIP proteins and corresponding sequences were acquired from the annotation profiles. The potential TdZIPs were further examined with the Conserved Domain Search (https://www.ncbi.nlm.nih.gov/Structure/bwrpsb/bwrpsb.cgi/, accessed on 10 January 2022). In addition, ZIP protein sequences from rice, maize, and Arabidopsis were downloaded from Ensembl Plant (http://plants.ensembl.org/index.html/, accessed on 10 January 2022) [54]. All ZIPs were aligned with clustalW (https://www.ebi.ac.uk/Tools/msa/clustalo/, accessed on 10 January 2022) [55], and the resulting alignments were used to construct a phylogenetic tree using the neighbor-joining method with 1000 bootstrap replications and MEGAX software as well as iTOL (https://itol.embl.de/upload.cgi/, accessed on 10 January 2022) [56]. The comparative synteny analysis of ZIPs between wild emmer wheat genome and common wheat CS genome (https://urgi.versailles.inra.fr/blast_iwgsc/, accessed on 10 January 2022) was performed using TBtool software v1.087 [57]. The *TdZIP* genomic sequences and CDS sequences of wild emmer wheat retrieved from Ensembl Plant database were compared using the gene structure display server program (GSGD, http://gsds.gao-lab.org/, accessed on 10 January 2022) to construct the exon/intron organization of ZIP genes [58]. All the *TdZIP* genes were mapped to wild emmer wheat chromosomes on the basis of physical location information from the database of wild emmer genome (WEWSeq_v.1.0). Conserved protein motifs were predicted using the MEME Suite web server (https://meme-suite.org/meme/tools/meme, accessed on 10 January 2022), with the maximum number of motifs specified as 15 and the optimum width of motif sets specified as 5 to 200 amino acids [59]. The online ExPASy (https://www.expasy.org/, accessed on 10 January 2022) and SMART sever (http://smart.embl.de/smart/batch.pl, accessed on 10 January 2022) were used to predict the amino acid length, transmembrane (TM) domain, and theoretical isoelectric point (PI) of the *TdZIP* proteins.

### 4.2. Gene Annotation and Cis-Element Analysis of TdZIPs

The TdZIP genes were annotated using Gene Ontology (GO) (http://www.geneontology.org/, accessed on 10 January 2022), Clusters of Eukaryotic Groups (KOG) (ftp://ftp.ncbi.nih.gov/pub/COG/KOG, accessed on 10 January 2022), and Swiss_Prot_annotation databases (https://ftp.ebi.ac.uk/pub/databases/swissprot, accessed on 10 January 2022). Cis-element analysis on the promoter regions (2000 bp) upstream of the translation initiation site (ATG) of TdZIP genes was performed using PlantCARE [60]. Microsoft Excel and TBtools were used to conduct statistical analysis and data visualization [57].

### 4.3. Plant Materials and Zn-Deficient Treatments

The advanced wheat line BAd7-209 (*T. aestivum* CN16/*T. dicoccoides* D97 F_12_) [61] and transgenic rice (*Oryza. Sativa* L. spp. *Japonica*) were used in the present study. The seeds were germinated on sterile culture dish with filter paper for 7 days, and seedlings were transplanted into Hogland nutrient solution with two concentrations of Zn, supplemented in the form of ZnSO_4_, including no Zn (Zn deficiency) and normal Zn supply (8.6 mg/L ZnSO_4_.7H_2_O). Leaves and roots of the individuals were harvested at 1, 3, 5, 7, and 9 days after Zn-deficient treatments. All samples were stored at −80 °C for further analysis.

### 4.4. RNA Extraction and Quantitative Real-Time PCR (qRT-PCR)

Total RNA from leaf and root samples was isolated using plant extraction kit v1.5 (Biofit Biotechnologies, Chengdu, China). First-strand cDNA synthesis was performed using the TaKaRa PrimeScriptTMRT Reagent Kit (Takara, Dalian, China) according to the manufacturer’s instructions. Quantitative real-time PCR (qRT-PCR) was performed using the Bio-Rad CFX96 Real-Time PCR System (Bio-Rad, Hercules, CA, USA) in 10 µL reaction volume containing 2 ng of cDNA, 5 µL of 1 × SYBR Premix Ex Taq (TaKaRa), and 0.5 µL (300 nM) of each primer. The GAPDH gene was used as the internal reference. The gene expression was quantified using the 2^−ΔΔCT^ method [62]. Three biological replicates were used for each data point.

### 4.5. Yeast Complementation Assay

The yeast strains wild-type DY1457 and *zrt1*/*zrt2* (DY1457 + zrt1::LEU2, zrt2::HIS3) mutant were used in this study. Yeast expression vector was pYES2, which has restriction enzyme cutting sites *BamHI* and *EcoRI*. The *TdZIP**6B-1* gene was connected to pYES2 using ClonExpress II One Step Cloning Kit (Vazyme Biotech Co., Ltd., Nanjing, China) and then transformed into *Escherichia coli* DH5α. The pYES2*-TdZIP6B-1* vector was further transformed into yeast strains using EX-Yeast Transformation Kit (ZOMANBIO). Yeast cells were grown in 1% yeast extract, 2% peptone, and 2% glucose (YPD). Low-Zn media (50 µM ZnSO_4_) and control media (1 mM ZnSO_4_) were prepared as described by [25]. The pH value was adjusted to 4.2 to enhance the EDTA affinity for Zn. The SD/-Ura liquid medium with galactose was used to select pYES2-TdZIP transformants. The positive transformants were incubated in SD/-ura liquid medium at 30 °C for 4 h. Yeast liquid culture was adjusted to OD_600_ value of 0.5 using SD-glucose/-ura/-Zn solution. A 1:10 dilution of the culture (1, 10^−1^, 10^−2^, 10^−3^, and 10^−4^) was made and 3 µL dilution of each was dropped onto different media for 3 days incubation at 30 °C.

### 4.6. Rice Transformation

The cDNA of *TdZIP6B-1* was cloned into the overexpression vector pCAMBIA2300-GFP (pCAMBIA2300-GFP-*TdZIP6B-1*). The construct had *KpnI* and *SpeI* on the 3′ side of the CaMV 35S promoter. An *Agrobacterium tumefaciens* strain (AGL1) carrying the above construct was used to transform rice (*Oryza. Sativa* L. spp. *japonica*) following the method of [11]. The T_1_ seeds obtained from the transformants were germinated on MS medium containing 50 mg L^−1^ hygromycin to select resistant plants. In addition, the hygromycin-resistant lines were further confirmed by PCR using gene-specific primer and qRT-PCR to detect the expression of *TdZIP6B-1* in transgenic lines. The *OsActin* gene was used as the internal reference. Homozygous T_3_ transgenic lines were selected for subsequent experimental analysis.

### 4.7. Measurement of Metal Concentration

The roots and shoots were sampled and dried at 70 °C for 3 days. The samples were wet-ashed by HNO_3_ (60%) as described previously [63]. After dilution, the Zn, Fe, and Mn concentrations were determined by inductively coupled plasma atomic emission spectrometry (SPS1200VR; Seiko, Tokyo, Japan) [52]

## 5. Conclusions

Zn homeostasis is important for plant development and adaptation to diverse stresses. The plant ZIP family proteins play critical role in uptake and transport of Zn ion. In this study, a genome-wide analysis of ZIP family gene in wild emmer was performed. A total of 33 *TdZIPs* genes were identified and compared with CS *TaZIP* genes. We found large sequence differences for most of them, and that *TdZIP5B-1* could not match *TaZIP* genes. We performed the phylogenetic, gene structure, chromosomal localization, expression, conserved motif, gene annotation (GO, KOG, and Swiss_Prot_annotation), cis-element, qRT-PCR, and functional analyses of *TdZIP* genes. *TdZIP1A-2*, *TdZIP1A-3*, *TdZIP6B-1*, and *TdZIP7B-2* have the ability to rescue the growth defects of the *zrt1*/*zrt2*Δ mutant. The overexpression of *TdZIP6B-1* in rice improved the Zn, Fe, and Mn concentrations in roots under normal Zn condition. Our results demonstrate that wild emmer can sever as a valuable reservoir of genetic variation in the study of Zn transporters and lay a significant foundation for future functional analysis and genetic improvement of Zn deficiency tolerance in wheat and *TdZIP6B-1* is a candidate Zn transporter that responsible for absorption and probably root-to-shoot translocation of Zn.

## Figures and Tables

**Figure 1 ijms-23-02866-f001:**
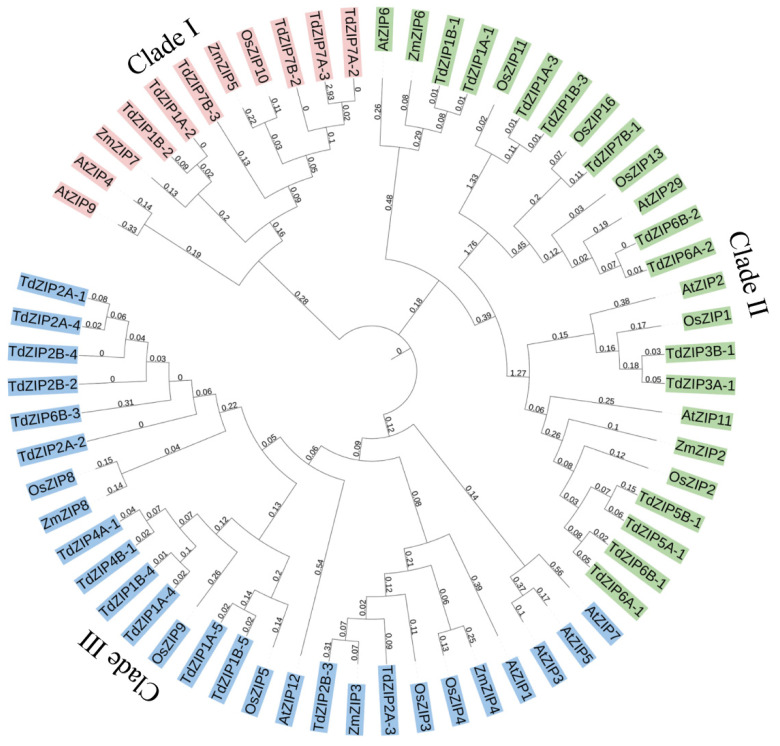
Phylogenetic relationships of the 33 *TdZIPs*, 11 *OsZIPs*, 7 *ZmZIPs*, and 11 *AtZIPs*. The subgroups (clades I, II, and III) of the ZIPs were marked in different color.

**Figure 2 ijms-23-02866-f002:**
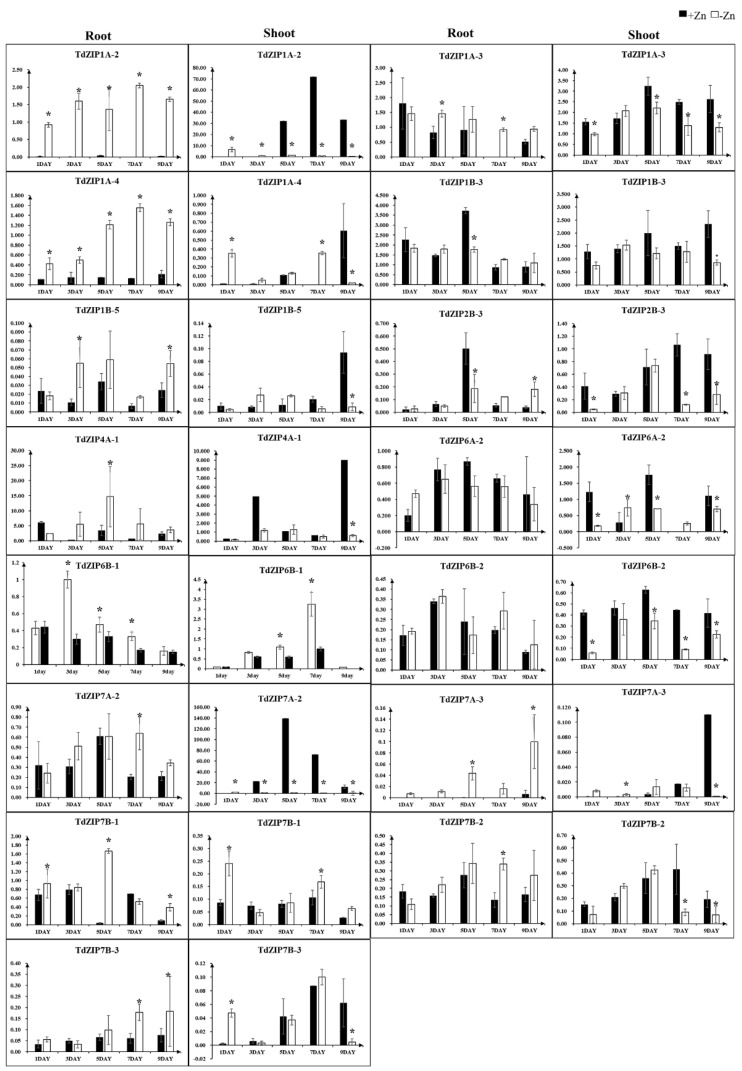
Expression levels of 15 *TdZIP* genes in the shoots and roots in response to Zn-deficient stress. Error bars indicate the mean values between three replicates ± standard deviation (SD). * denotes the statistically differences at *p* < 0.05 (Student’s *t*-test).

**Figure 3 ijms-23-02866-f003:**
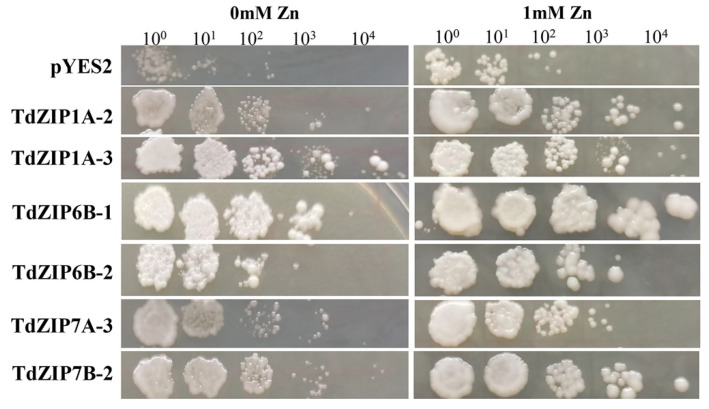
Complementation of yeast Zn uptake mutant (*zrt1*/*zrt2*Δ) with *TdZIP* genes under different Zn concentrations. The yeast *zrt1*/*zrt2*Δ mutant transformed with the empty vector *pYES2* was used as a negative control. Each spot represents a 1:10 dilution of the culture starting with an OD of 0.5 on the far left (10-, 100-, 1000-, 10,000-fold dilutions).

**Figure 4 ijms-23-02866-f004:**
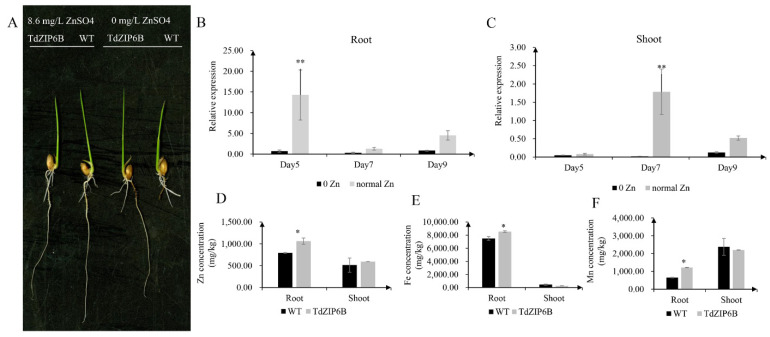
Comparison of phenotypes and metal concentration between *TdZIP6B-1* overexpression lines (TdZIP6B) and WT plants under no Zn (0 mg/L ZnSO_4_) and normal Zn (8.6 mg/L ZnSO_4_.7H_2_O) conditions. (**A**) Morphology of rice seedlings that were exposed to no Zn and normal Zn for 14 days. (**B**,**C**) The expression levels of *TdZIP6B-1* in transgenic lines exposed to no Zn and normal Zn conditions for 5, 7, and 9 days. (**D**–**F**) The Zn, Fe, and Mn concentrations in roots and shoots of TdZIP6B and WT plants under normal Zn condition at 14 days. Error bars show SE and the symbol * and ** indicate statistical differences (*p* < 0.05 and *p* < 0.01, respectively).

**Figure 5 ijms-23-02866-f005:**
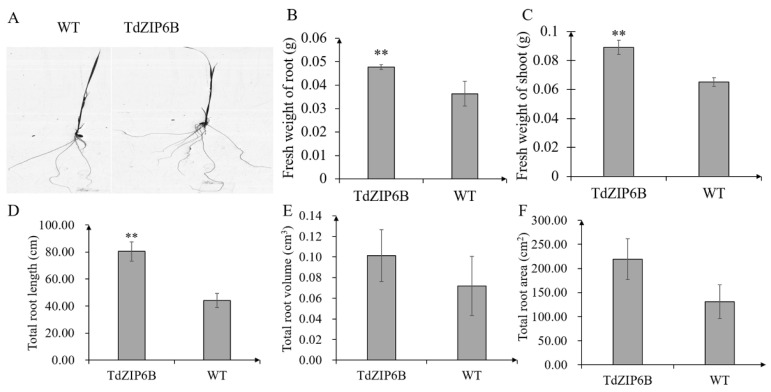
Seedling phenotypes of *TdZIP6B**-1* overexpression lines (TdZIP6B) and WT plants exposed to normal Zn condition at 14 days. (**A**) Root phenotype; (**B**,**C**) fresh weight of roots and shoots; (**D**–**F**) total roots length, volume, and area. Error bars show SE, and the symbol ** indicates statistical differences (*p* < 0.01).

**Table 1 ijms-23-02866-t001:** Information and physicochemical characteristics of the *TdZIP* genes.

Gene	Sequence ID	Chromosome	Protein Length (Amino Acids)	Transmembrane Domain	Isoelectric Point (PI)	Subcellular Location
*TdZIP1A-1*	*TRIDC1AG014690.1*	1A	376	8	6.79	Plasma membrane
*TdZIP1A-2*	*TRIDC1AG018050.1*	1A	148	3	7.83	Plasma membrane
*TdZIP1A-3*	*TRIDC1AG019100.1*	1A	577	13	6.85	Plasma membrane
*TdZIP1A-4*	*TRIDC1AG044090.1*	1A	371	9	6.26	Plasma membrane
*TdZIP1A-5*	*TRIDC1AG044100.1*	1A	351	7	6.55	Plasma membrane
*TdZIP1B-1*	*TRIDC1BG018820.1*	1B	395	8	6.21	Plasma membrane
*TdZIP1B-2*	*TRIDC1BG022010.1*	1B	119	3	6.91	Plasma membrane
*TdZIP1B-3*	*TRIDC1BG023900.1*	1B	577	13	7.54	Plasma membrane
*TdZIP1B-4*	*TRIDC1BG049910.1*	1B	359	9	6.38	Plasma membrane
*TdZIP1B-5*	*TRIDC1BG049920.3*	1B	354	7	6.36	Plasma membrane
*TdZIP2A-1*	*TRIDC2AG017840.1*	2A	115	3	7.78	Plasma membrane
*TdZIP2A-2*	*TRIDC2AG017850.1*	2A	68	1	5.05	Plasma membrane
*TdZIP2A-3*	*TRIDC2AG061130.1*	2A	167	5	9.8	Plasma membrane
*TdZIP2A-4*	*TRIDC2AG071150.1*	2A	104	3	9.3	Plasma membrane
*TdZIP2B-2*	*TRIDC2BG021610.1*	2B	170	4	6.33	Plasma membrane
*TdZIP2B-3*	*TRIDC2BG064770.1*	2B	260	3	6.89	Plasma membrane
*TdZIP2B-4*	*TRIDC2BG077000.1*	2B	361	7	6.1	Plasma membrane
*TdZIP3A-1*	*TRIDC3AG074590.1*	3A	383	9	5.83	Plasma membrane
*TdZIP3B-1*	*TRIDC3BG085920.1*	3B	355	9	6.59	Plasma membrane
*TdZIP4A-1*	*TRIDC4AG003470.1*	4A	206	4	6.64	Plasma membrane
*TdZIP4B-1*	*TRIDC4BG048110.1*	4B	365	7	6.36	Plasma membrane
*TdZIP5A-1*	*TRIDC5AG039960.1*	5A	188	5	5.2	Plasma membrane
*TdZIP5B-1*	*TRIDC5BG042350.1*	5B	349	7	9.21	Plasma membrane
*TdZIP6A-1*	*TRIDC6AG012920.1*	6A	358	9	5.41	Plasma membrane
*TdZIP6A-2*	*TRIDC6AG022380.1*	6A	276	8	8.73	Endoplasmic reticulum
*TdZIP6B-1*	*TRIDC6BG018040.1*	6B	364	9	5.28	Plasma membrane
*TdZIP6B-2*	*TRIDC6BG028970.1*	6B	229	6	8.76	Endoplasmic reticulum
*TdZIP6B-3*	*TRIDC6BG030420.1*	6B	382	9	5.81	Plasma membrane
*TdZIP7A-2*	*TRIDC7AG050460.2*	7A	366	6	6.13	Plasma membrane
*TdZIP7A-3*	*TRIDC7AG058500.1*	7A	250	3	6.21	Plasma membrane
*TdZIP7B-1*	*TRIDC7BG041340.4*	7B	170	5	5.3	Endoplasmic reticulum
*TdZIP7B-2*	*TRIDC7BG043370.1*	7B	221	3	6.1	Plasma membrane
*TdZIP7B-3*	*TRIDC7BG051280.1*	7B	162	2	6.18	Plasma membrane

## Data Availability

The data presented in this study are available in the article and Appendix A.

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
