# Peer review of "Genome-Wide Investigation and Functional Verification of the ZIP Family Transporters in Wild Emmer Wheat"

_ijms, 2022, doi:10.3390/ijms23052866_

Round 1
Reviewer 1 Report
The research paper by Gong et al. performed a genome-wide analysis of ZIP genes in emmer wheat. They also performed phylogenetic analysis, cis-element, gene expression analysis and Yeast complementation analysis. However, I do have some concerns before its final acceptance.
- Materials and method section, please provide Pfam ID used for ZIP family transporters identification.
- Line 101-103 “Most of TdZIP proteins were predicted ……localized on the endoplasmic reticulum”. In earlier studies it was reported that ZIPs were located on the plasma membrane. Why three ZIPs were localized on ER, authors should explain a possible reason for this.
- In a phylogenetic analysis of ZIP family transporters, authors had done analysis using protein sequences from emmer wheat, rice and A. thaliana only. They can draw a better picture/result if they use more species for phylogenetic analysis.
- The results of RNAseq should be provided in more detail. Is there any correlation found between qPCR and RNAseq data?
- How many biological replicates were used in qRT-PCR analysis. This information was not provided.
- Figure S1, a scale for chromosome in Mb should be provided.
Author Response
Response to Referees Letter
Thank you, we have fully addressed all of the concerns raised by you, point by point and we have edited the text as specified in “Response to Referees Letter”. The reviewer/editor's comments are listed below and marked in red Italics, response to the reviewer/editor's comments are marked below in black, while the edited/revised/new text from the paper is marked below in blue.
Response to Reviewer 1 Comments
Q1:
Materials and method section, please provide Pfam ID used for ZIP family transporters identification.
Response: Thank you. We have provided Pfam ID used for ZIP family transporters identification.
“The ZIP DNA-binding domain (PF02535.24) downloaded from Pfam protein families database (http://pfam.xfam.org/) was used to identify ZIP genes from the wild emmer genome using HMMER 3.0.”
Q2:
Line 101-103 “Most of TdZIP proteins were predicted ……localized on the endoplasmic reticulum”. In earlier studies it was reported that ZIPs were located on the plasma membrane. Why three ZIPs were localized on ER, authors should explain a possible reason for this.
Response: Thank you. Previous reports showed that some ZIP family proteins had an endoplasmic reticulum subcellular localization (Durmaz et al. 2011, Plant Molecular Biology Reporter, 29, 582-596; Xu et al. 2010, Functional Plant Biology, 2010, 37, 194-205). We used software ProtComp 9.0 (http://linux1.softberry.com/berry.phtml?topic=protcomppl&group=programs&subgroup=proloc) to predict the subcellular location of ZIP proteins and three ZIPs were localized on ER. We assume the possible reason is that the three ZIP genes are responsible for transporting zinc from the ER to the cytoplasm.
Revised in the MS: “Most of TdZIP proteins were predicted to be localized on the plasma membrane except three TdZIPs (TdZIP6A-2, TdZIP6B-2, and TdZIP7B-1) were predicted to be localized on the endoplasmic reticulum (Table 1), suggesting that these three genes may be resposible for transporting Zn from the ER to the cytoplasm.”
Q3:
In a phylogenetic analysis of ZIP family transporters, authors had done analysis using protein sequences from emmer wheat, rice and A. thaliana only. They can draw a better picture/result if they use more species for phylogenetic analysis.
Response: Thank you. We re-drew a better picture used wild emmer, rice, maize and A. thaliana for phylogenetic analysis. We chose these species because there are many reports related to ZIP transporters in these species, the phylogenetic analysis will help us to understand the potential role the TdZIP genes in our study.
We have modified the corresponding parts (line 110-115 and line 255-270).
Q4:
The results of RNAseq should be provided in more detail. Is there any correlation found between qPCR and RNAseq data?
Response: Thank you. The RNA-seq data were adopted from wild emmer genome (http://202.194.139.32/expression/emmer.html). We have provided more details for the RNAseq data.
Results section: “To understand the potential expression pattern of TdZIP genes in different tissues, the expressions of 33 TaZIPs in different tissues sampled at different time points were retrieved from public available RNA-seq data of wild emmer (Avni et al., 2017). The log2 (TPM+1) value was used for the heat map display (Fig. S6). We found TdZIP genes showed different expression patterns. According to the expression data, fourteen TdZIP genes including TdZIP1A-1, TdZIP1A-2, TdZIP1A-3, TdZIP1B-3, TdZIP1B-5, TdZIP2A-2, TdZIP2A-3, TdZIP2B-3, TdZIP6A-2, TdZIP6B-1, TdZIP6B-2, TdZIP6B-3, TdZIP7A-2, and TdZIP7B-2 were highly expressed in different tissues sampled at different time points. Some genes were expressed in all sampled tissues. For example, TdZIP6B-2 had high expression in root, leaf, developing spike, lemma, glume, flower, and grain at most time points. TdZIP1A-1 was highly expressed in leaf, lemma, glume, and grain. In addition, some genes only expressed at specific developmental stages of specific tissues. For example,”
Since the RNAseq data and the qPCR results are collected from different wild emmer genotypes and growth on different treatment, we did not find any obvious correlation between these two data.
Q5:
How many biological replicates were used in qRT-PCR analysis. This information was not provided.
Response: Thank you. The result of qRT-PCR analysis had three biological replicates. We had provided this information in material and methods section.
“Three biological replicates were used for each data point.”
Q6:
Figure S1, a scale for chromosome in Mb should be provided.
Response: Thank you. We have provided a scale for chromosome in Fig S1.

Reviewer 2 Report
The manuscript titled “Genome wide investigation and functional verification of the ZIP family transporters in wild emmer wheat” is a well-planned study and should be accepted for publication after some changes, as listed below:
- Please italicized the names of the genes throughout the manuscript
- Improve the resolution of all the figures, especially Figure 2
- What is the control for the analysis for the expression pattern of the ZIP genes?
- Why 1-100000-fold dilution was not tried in complementation experiment?
Author Response
Response to Referees Letter
Thank you, we have fully addressed all of the concerns raised by you, point by point and we have edited the text as specified in “Response to Referees Letter”. The reviewer/editor's comments are listed below and marked in red Italics, response to the reviewer/editor's comments are marked below in black, while the edited/revised/new text from the paper is marked below in blue.
Response to Reviewer 2 Comments
Q1:
Please italicized the names of the genes throughout the manuscript.
Response: Thank you. We have italicized the names of the genes throughout the manuscript.
Q2:
Improve the resolution of all the figures, especially Figure 2.
Response: Thank you. We have improved the resolution of all the figures.
Q3:
What is the control for the analysis for the expression pattern of the ZIP genes?
Response: The aim of the expression analysis was to reveal the transcript changes of TdZIPs in response to different concentrations of Zinc treatments, we used same wheat lines and two concentrations of Zn for the expression analysis. The normal Zn concentration treatment was considered as control compared to Zn-deficient condition.
Q4:
Why 1-100000-fold dilution was not tried in complementation experiment?
Response: In many previous reported yeast complementary experiments, the dilution was 1-10000-fold (Weber et al. 2004; Milner et al. 2013) that is enough to see the significant difference between the experimental group and the control group at 10000-fold dilution and it works for our experiment. So, we didn’t try 100000-fold dilution in complementation experiment.
Weber, M.; Harada, E.; Vess, C.; Edda v. Roepenack‐Lahaye,; Clemens, S. Comparative microarray analysis of Arabidopsis thaliana and Arabidopsis halleri roots identifies nicotianamine synthase, a ZIP transporter and other genes as potential metal hyperaccumulation factors. The Plant Journal. 2004, 37, 269-281.
Milner, M.; Seamon, J.; Craft, E.; Kochain, L. Transport properties of members of the ZIP family in plants and their role in Zn and Mn homeostasis. Journal of Experimental Botany. 2013, 64, 369-381.

Round 2
Reviewer 1 Report
The authors have clarified most of the questions I raised in my previous review. Now I feel the manuscript could be accepted for publication.